# Comparative Study on Gas-Sensing Properties of 2D (MoS_2,_ WS_2_)/PANI Nanocomposites-Based Sensor

**DOI:** 10.3390/nano12244423

**Published:** 2022-12-11

**Authors:** Hemalatha Parangusan, Jolly Bhadra, Razen Amer Al-Qudah, Elhassen Cheikh Elhadrami, Noora Jabor Al-Thani

**Affiliations:** 1Qatar University Young Scientists Center (QUYSC), Qatar University, Doha P.O. Box 2713, Qatar; 2Center for Advanced Materials, Qatar University, Doha P.O. Box 2713, Qatar; 3Department of Chemical Engineering, Qatar University, Doha P.O. Box 2713, Qatar

**Keywords:** transition-metal dichalcogenides, polyaniline, conducting polymer, gas sensor, nanocomposites

## Abstract

NH_3_ is a highly harmful gas; when inhaled at levels that are too high for comfort, it is very dangerous to human health. One of the challenging tasks in research is developing ammonia sensors that operate at room temperature. In this study, we proposed a new design of an NH_3_ gas sensor that was comprised of two-dimensional (TMDs, mainly WS_2_ and MoS_2_) and PANI. The 2D-TMDs metal was successfully incorporated into the PANI lattice based on the results of XRD and SEM. The elemental EDX analysis results indicated that C, N, O, W, S and Mo were found in the composite samples. The bandgap of the materials decreased due to the addition of MoS_2_ and WS_2_. We also analyzed its structural, optical and morphological properties. When compared to MoS_2_ and PANI, the proposed NH_3_ sensor with the WS_2_ composite was found to have high sensitivity. The composite films also exhibited response and recovery times of 10/16 and 14/16 s. Therefore, the composite PANI/2D-TMDs is a suitable material for NH_3_ gas detection applications.

## 1. Introduction

One of the most common air pollutants that significantly affects the environment is ammonia (NH_3_), and there has recently been a great demand for monitoring and regulating this contaminant. It is a highly hazardous gas that is utilized in a variety of commercial and residential applications, including the compound fertilizer, synthetic fiber, biodiesel, energy plants’ and dye industry [1,2,3]. Additionally, exposure to excessive levels of ammonia gas can have harmful impacts on a human’s body, including eye irritation, the skin, throat and respiratory system [4,5]. Hence, exposed gas detection has a significant impact on environmental, health and safety issues. Therefore, it is essential to develop ammonia gas sensors for process and safety control in order to detect environmental pollution. Ammonia can be detected using a variety of monitoring techniques, including electrochemical methods, optical spectroscopy and gas chromatography–mass spectrometry [6,7,8]. In comparison to other sensing techniques, metal-oxide-based electrochemical approaches are more attractive because they can be integrated into portable sensors and have easy experimental procedures, quick reaction times and good stability. However, for typical metal-oxide-based sensors, such as TiO_2_, ZnO and SnO_2_, a high temperature is required to activate gas adsorption. The high operating temperature probably releases explosive or flammable analytics in addition to decreasing the sensor’s long-term stability [9,10]. Hence, it is crucial to develop sensors that can operate at room temperature.

As an alternative to metal oxide semiconductors for smart sensing applications, conducting polymers (CPs) have attracted more attention. This is because of their ability to operate at room temperature, their quick reaction and recovery periods, their low cost of production, how easily they can be deposited on a variety of substrates and their extensive chemical abilities for structural modification. Due to its simpler synthesis, high environmental stability and great electrical and electrochromic properties, PANI, in particular, has developed to be among the conducting polymers with the greatest technological significance [11]. It has been utilized extensively in gas-sensing and organic solar cell applications. One effective way to increase the detecting performance of gas sensors at room temperature is to fabricate PANI composites with 2D materials.

2D-TMDs (transition-metal dichalcogenides) have been shown to be attractive materials for gas-sensing applications because of their superior electrical conductivity, high surface-to-volume ratio and sensitive surfaces [12,13]. Both WS_2_ and MoS_2_ have recently demonstrated their promise and made significant advancements in the field of toxic-gas-sensing technologies [14]. TMDs materials are layered substances with strong covalent interlayer connections and weak interlayer van der Waals interactions. Each layer in TMDs is made up of three atomic planes: a triangle-shaped lattice of transition-metal atoms sandwiched between two triangle-shaped lattices of chalcogen atoms. This makes TMDs strong candidates for advanced electrical devices such as energy storage, optoelectronics and sensors as proof-of-concept devices [15,16]. To overcome some specific issues with conventional sensing materials, such as slow sensing response and poor selectivity, TMDs have been used as efficient chemical sensors [17]. TMDs have a variety of desirable features, such as variable nanoscale layer thickness, a high surface-to-volume ratio, ease of surface functionalization and high compatibility for device integration. These features are all essential for creating high-performance sensors for use in various analytics [18]. Keerthi et al. [19] prepared a hybrid heterostructure containing a La-MoS_2_-based gas sensor. When La was added to MoS_2_, its sensitivity towards toxic gases and volatile organic chemicals significantly increased. Using a 2D heterostructure, Cho et al. [20] developed a new chemical sensor that paves the way for an important sensing platform for wearable electronics. Recent research on WS_2_, MoS_2_ and their hybrid structures (functionalized nanofibers, Quantum dots and metal-doped nanoflower structures [21,22]) in gas-sensing applications shows significant sensitivity at low operating temperatures with particular selectivity to certain gases such as H_2_S and NH_3_ [23]. The lower operating temperature of sensors based on transition-metal chalcogenides. Because they typically have a lower bandgap and better conductivity, the oxides of the corresponding metals perform better than those oxides. Additionally, similar to semiconducting metal oxide sensors, the gas selectivity of TMDS is strongly associated to the affinities of surfaces for adsorbing various analytics and related surface charging/polarization effects. Additionally, it has been found that reversible doping of the chalcogenide lattice with heteroatoms, in addition to surface charging/polarization, can greatly contribute to sensing [24].

In this context, the development of an NH_3_ sensor using a composite PANI/2D-TMDs is the main focus of this research. We propose an easy, reliable, simple and low-cost material for the detection of gas. Furthermore, we investigate the gas-sensing performance of pure PANI and its nanocomposites. The prepared PANI/2D-TMDs composites are also analyzed to find their structural, morphological, optical and NH_3_-sensing properties using XRD, FTIR, SEM, TEM, UV-vis and sensing studies. The composite films were coated on ITO electrodes by the spin-coating method and tested for NH_3_ detection.

## 2. Materials and Methods

Aniline, Ammonium persulfate and HCL were purchased from Sigma-Aldrich. Molybdenum disulfide (MoS_2_) and Tungsten disulfide (WS_2_) powders were used for synthesis. All the chemicals were derived from Sigma-Aldrich.

### 2.1. Liquid-Exfoliated MoS_2_ and WS_2_ Sheets

Both MoS_2_ and WS_2_ (30 mg) powders and 50 mL of DMF were added to a 100 mL glass container, separately, and the bath sonication process was used to prepare the powders for dispersion. The mixture was processed in the sonication bath for 12 h because most of the un-exfoliated powders tended to settle at the bottom of the glass container. The mixture was then separated into the stable MoS_2_ and WS_2_ dispersion and the un-exfoliated sediments using a centrifuge at 4000 rpm for 10 min.

### 2.2. Synthesis of Liquid-Exfoliated Sheet/PANI Composites

The typical preparation procedure of pure PANI and its composites was as follows (Figure 1): 0.2 M aniline was added into 50 mL of 0.01 M Hydrochloric acid solution, and it was agitated in an ice bath for 2 h. The nanosheet dispersion was dropped into the aniline solution under vigorous stirring. The amount of dropped nanosheet dispersion varied in the range of 5–15 mL. The resulting suspension was stirred for 1 h. Ammonium persulfate (0.2 M), which served as the oxidant in this solution, was added dropwise while stirring continuously for a whole night. The precipitation was then filtered and washed with ethanol and DI water and vacuum oven dried at 60 °C for 6 h. The same process was used to synthesize pure PANI without the incorporation of nanosheets.

### 2.3. Materials’ Characterization

The prepared samples’ structural behavior was analyzed through an X-ray diffractometer (Empyrean, Panalytical, UK) with the PerkinElmer Spectrum 400 spectrophotometer. Fourier infrared transform (FTIR) spectrum was recorded in the 400–4000 cm^−1^ range. The surface morphologies of the samples were examined by using scanning electron microscopy (Nova Nano SEM 650) and transmission electron microscopy (JEOL JEM-3010). The gas sensor device fabrication and the testing system were similar to our previous report [25]. Re-patterned interdigitated ITO (Indium tin oxide/glass substrates) were purchased from Ossila for the fabrication of the sensor. The thickness of ITO on the glass substrate was 100 nm. Each substrate, with dimensions (20 mm × 15 mm), consisted of the five ITO-based interdigitated sensation electrodes, while each interdigitated sensation electrode was comprised of three channels with a size of 30 mm × 50 µm. The ITO substrate was cleaned sequentially in acetone and deionized water in an ultrasonic bath for 10 min before coating with the sensing materials.

Finally, the surface of the ITO was dried by nitrogen blow. The pure and composite powder (200 mg) was first dissolved in 2 mL of ethanol before being used to deposit the sensor film. Then, the solution was spin-coated onto the substrate at 5000 rpm for 60 s and dried at 60 °C for 1 h. Our group already optimized the spinning time and spinning velocity of the nanocomposite-based sensing film. The thickness of the composite films was recorded using profilometer (LEICA DCM8), and the average thickness was found to be 390 nm. Gas-sensing properties were tested by using an assembled experimental set up. The set up included Keithley 2400, source meter, measuring chamber and measuring devices. The Keithley 2400 measured the electrical properties of the fabricated sensors. Figure 2 shows the experimental set up for sensor measurement. The gas response, S, is defined as S = R_g_/R_a_, where R_g_ is the resistance of the sensor in the presence of gas. R_a_ is the resistance of the sensor in the presence of air.

## 3. Results and Discussion

### 3.1. Structural Analysis

The structural properties of the PANI/MoS_2_ and PANI/WS_2_ composites were determined through XRD analysis, and their diffraction patterns are shown in Figure 3. Figure 3a displays the diffraction pattern for the PANI and PANI/MoS_2_ samples. The diffraction peaks at 20.5° and 25.3° correspond to (020) and (200) pure PANI crystalline planes, respectively. The diffraction peak at 25.3° could be attributed to PANI chains that are periodically perpendicular [26]. It is evident from the XRD pattern of the composite samples shown in Figure 3a that the main characteristic diffraction peaks of MoS_2_ were also found. The result demonstrates that the PANI/MoS_2_ nanocomposite were synthesized successfully. Similarly, the formation of PANI/WS_2_ composites was confirmed by using XRD patterns, and it is shown in Figure 3b.

The crystallite sizes of pure PANI and its composite samples were calculated using the Scherrer formula from the broadening of the diffraction peak of the (020) plane, and the values are listed in Table 1.

The Debye–Scherrer equation can be used to calculate the crystallite size (*D*)
(1)D=(0.9λ)/βcosθ
where λ is the wavelength of the incident X-ray (1.5406 Å), β is the full width at half maximum (FWHM) and *θ* is the diffraction angle. The formation of new nucleation centers by 2D-TMDs ions could be related to the reduction in crystallite size that occurs when 2D-TMDs are introduced into the PANI lattice [27]. Most notably, XRD data demonstrate that while the incorporation of 2D-TMDs had an effect on the PANI lattice, it had no negative effects on phase segregation, which may be a drawback for sensing applications.

The relation between the microstrain and the number of defects is directly proportional, and the following relation can be used to estimate the microstrain,
(2)ε=βcosθ4

The calculated values of ε are shown in Table 1, and it is clear from the values that the microstrain increases as the 2D ion concentration increases. This, in turn, demonstrates that the lattice defects increase with the 2D ion concentration.

The dislocation density (δ) is calculated using the crystallite size (*D*), and its values are shown in Table 1 along with the following relation:(3)δ=1D2

The structural parameters are shown in Table 1. There was a slight difference in the structural properties of the PANI/2D-TMDs composites compared to pure PANI, which can be attributed to the 2D atoms occupying different sites in the PANI lattice and the number of defect formations in PANI, which is favored for better gas sensing. These findings provide more evidence that, due to the increasing defects, the PANI/TMDs composites’ crystallite size decreased.

The chemical composition of and the interaction between pure PANI and its nanocomposites were analyzed by using FTIR spectroscopy. Figure 4a,b display the PANI and PANI/2D-TMDs hybrid FTIR spectra.

Peaks at 1461 cm^−1^ and 1551 cm^−1^ in the FTIR spectra of pure PANI correspond to the C=C and C=N and the vibrations of the benzenoid and quinoid rings, respectively. The plane bending vibrations -C-H could be the reason for the peak at 1289 cm^−1^. The peak at 1132 cm^−1^ suggests the development of a doped polymer [28]. The major peaks for both PANI and 2D-TMDs could be found in the FTIR range of the composite samples. These findings suggest that polyaniline and the matrix of the 2D material interact at the molecular level. In Figure 4a, the peaks at 604 cm^−1^ and 940 cm^−1^ are associated with the components Mo-S and S-S, respectively. The band positioned at 599 cm^−1^ is attributed to the W-S bond (Figure 4b).

### 3.2. Morphological Analysis

FE- SEM and TEM analyses were used to study the surface morphology of pure PANI and its nanocomposites. Figure 5a shows a typical FESEM image of pure PANI, revealing a large number of agglomerated nanoparticles. The FESEM images of the MoS_2_ and WS_2_ samples are shown in Figure 5b,d. Pure MoS_2_ and WS_2_ nanosheets are suitable as a substrate for anchoring other nanoparticles because they are wide enough and have lateral sizes of about 6–7 µm. In Figure 5d, WS_2_ nanosheets exhibit some evident folding, and one can conclude that they are few-layer materials. Figure 5c,e show the morphological behavior of the composite samples. The results indicate that the FE-SEM images show the nanosheets decorated with PANI nanoparticles in the composite. In both composite materials, the sheets were interconnected with nanoparticles, which allows gas to move into the inside of the material and enhances NH_3_ gas sensing.

Figure 6a shows the EDX results of the composite materials. The EDX results reveal that the sample consisted of the elements carbon, oxygen and nitrogen, which confirms PANI formation. Figure 6b,c show that the EDX patterns of PANI and its nanocomposites reveal the presence of carbon, oxygen and nitrogen and also Mo, S, W and S, indicating the successful synthesis of PANI/2D-TMDs.

TEM images were utilized to further elucidate the morphology of PANI and its nanosheet composite, as shown in Figure 7.

A TEM image of pure PANI is shown in Figure 7a. It shows nanoparticles with less agglomeration. The morphology of pure MoS_2_ and WS_2_ is shown in Figure 7b,d, which exhibits a huge micrometer-sized uneven sheet that was densely stacked. The layered structure of MoS_2_, WS_2_ and PANI nanoparticles, as well as their distinct presence, are shown in Figure 7c,e. The results demonstrate that PANI successfully polymerized on the surface of MoS_2_ and WS_2_, which suggests the interaction between the negatively charged liquid-exfoliated sheets and the positively charged anilinium hydrochloride cations.

### 3.3. Optical Analysis

The optical characteristics are essential in determining how the nanocomposites affect the absorption properties of the material. Figure 8a,c show the UV-vis absorption spectra of pure PANI and its nanosheet composites. The UV-vis absorption spectra of the PANI/WS_2_ nanocomposites and pure PANI (inset) are shown in Figure 8c. A strong absorption band was observed at 300–400 nm. These were the findings of the typical benzenoid ring π-π transition and the expected charge transference that occur between benzenoid and quinoid [29]. It is important to note that, in comparison to pure samples, the absorption peaks of the composite samples significantly increased above this range. The absorption peaks of the composite samples also shifted toward the longer wavelength region. The interaction between the hydroxyl groups of TMDs and the quinoid ring of emeraldine salt may be the reason for this red shift because it makes it easier for charges to be transferred from highly reactive imine groups at the quinoid unit of emeraldine salt to TMDs through partial hydrogen bonding [30].

The following equation was used to calculate the material’s band gap energy (*E_g_*):(4)$${\left(\alpha h\nu \right)}^{n}=A\left(h\nu -E_{g}\right)$$

The band gap of the samples was determined using Tauc’s plots (Figure 8b,d). The obtained *E_g_* values of the samples are shown in Table 1, and it can be seen that *E_g_* decreases for the composite samples. In other words, *E_g_* exhibited a red shift as MoS_2_ and WS_2_ concentrations increased.

This is due to the lattice disorder caused by the substitution of MoS_2_ and WS_2_ ions with PANI ions, which results in defect energy levels below the conduction band edge and produces the red shift of the band gap energies [31]. The many-body processes that can decrease the band gap were demonstrated to be the result of electron interaction and impurity dispersion, which, in turn, led to the merging of an impurity band into the conduction band and, as a result, a decrease in the band gap.

### 3.4. Gas Sensing Studies

Among the several types of gas sensors are electrochemical gas sensors, contact combustion gas sensors and metal oxide semiconductor gas sensors. Conducting polymers (CPs)—one of them—have attracted more attention as an alternative solution to metal oxide semiconductors. This is because they operate at room temperature, with quick response and recovery times and low fabrication costs [32]. When a material absorbs certain gas molecules, its electrical characteristics can generally change. The concentration of dopants, the surface modification, crystallite size and chemical composition influence the sensing responses [33,34]. The operating temperature is a key indicator of how effectively sensors are working. The electrical resistance of PANI and its nanocomposites are studied as a function of ammonia (NH_3_) at room temperature in Figure 9a–d.

Figure 9b,d show the response to NH_3_ with concentrations ranging from 0.5 ppm to 2 ppm at room temperature. Due to the fact that the PANI film shows a p-type semiconductor characteristic when exposed to gas, the adsorbed NH_3_ gas molecules donate electrons to the PANI chains after adsorption, which results in a decrease in the carrier concentration of PANI and an increase in resistance. Compared with pure PANI and PANI/MoS_2_, the combination of PANI/WS_2_ hybrid film exhibited an improved response. As shown in Figure 9a,c, the response and recovery of the sensor to 2 ppm NH_3_ were about 31/34, 25/29, 22/27, 10/16, 22/23, 20/23 and 14/16 s, respectively. This shows that the response and recovery times of the sensor decreased as the MoS_2_ and WS_2_ concentration increased. In the narrowing bandgap materials, the substitution of MoS_2_ and WS_2_ ions into the PANI ions resulted in lattice disorder. The XRD and UV-vis spectra reveal that PANI/15 WS_2_ indicated more defects than the other materials. Consequently, the hybrid material significantly enhanced PANI’s gas-detecting capabilities.

The sensor’s long-term stability is a crucial factor in determining its practical application. The stability of the sensor towards 2 ppm NH_3_ at room temperature is shown in Figure 10a. It was observed that the gas response did not vary considerably over the duration of the measurement in 12 days, indicating the stability of the sensors based on PANI/MoS_2_ and PANI/WS_2_. The effect of humidity on the sensing performance of the sensor was investigated. At room temperature, the sensor’s response value to 2 ppm NH_3_ was tested for various relative humidity ranges of 10–70%. The result is presented in Figure 10b. The findings revealed that the response increased slightly as relative humidity increased. The results were attributed to the re-doping effect of water on acidified polyaniline. In addition, water can increase the rate at which PANI reacts with NH_3_.

Figure 11 shows the gas responses of the PANI, PANI/15 MoS_2_ and PANI/15 WS_2_ nanocomposite sensor towards ammonia, nitrogen dioxide, ethanol and acetone at room temperature, respectively. It was evident that the response to ammonia was significantly higher than that of the other tested gases, indicating that the nanocomposite sensor had strong selectivity for ammonia detection. One explanation could be that ammonia is more polarized than other gases because it has a single species lone electron pair.

According to the results, the NH_3_ molecule adsorption on the PANI/TMDs nanocomposite sensing film may be the reason for the resistance changing behaviors. When using a pure PANI sensor, the response is mainly due to protonation and deprotonation caused by the adsorption and desorption of NH_3_ gas molecules, respectively. Here, exposure to NH_3_ gas on the sensor surface caused the emeraldine salt form of PANI to change into the emeraldine base form. As a result, PANI’s resistance increased as its hole density decreased [35]. For hybrid films, gas sensing is primarily dependent on the trapping of NH_3_ gas molecules in the nanocomposite and the charge in the surface resistance of the nanocomposite when the gas adsorbs and reacts with it. This mechanism is in addition to the one discussed above. Exfoliated MoS_2_ and WS_2_ surfaces have several defects that could allow them to chemisorb ambient oxygen. Because NH_3_ is a donor molecule, it increases the resistance of the sensing layer by donating electrons to lower the concentration of holes in the sensing layer [36]. A comparison of the data showing the NH_3_ gas-sensing behavior of the present work with the literature reports on TMDs-based sensors is given in Table 2.

## 4. Conclusions

In this work, we prepared PANI and PANI/2D-TMDs by an in situ polymerization method, and the layered materials and transition-metal dichalcogenides MoS_2_ and WS_2_ were exfoliated and dispersed under sonication using DMF. The structural, morphological and compositional characteristics were successively studied. The XRD patterns of the samples indicate the formation of pure PANI and its nanocomposites. Defects in the lattice sites and the formation of new energy levels in the bandgap were due to the addition of MoS_2_ and WS_2_ to PANI. The surface morphology of the samples was investigated by TEM and SEM. It is concluded that the addition of 2D-TMDs has a significant impact on structural and optical properties, which also reveals a major impact on gas-sensing properties. A significant difference between the resistance values of the pure PANI and PANI/2D-TMDs was seen in the room-temperature sensing studies. It is revealed that the WS_2_-based NH_3_ sensor has higher sensitivity when compared to MoS_2_ and PANI. The proposed gas sensors have significant potential for use in industrial applications and the field of gas sensing.

## Figures and Tables

**Figure 1 nanomaterials-12-04423-f001:**
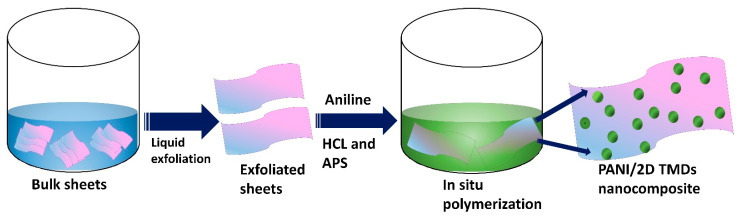
Schematic diagram of PANI and PANI/2D-TMDs composites by in situ polymerization process.

**Figure 2 nanomaterials-12-04423-f002:**
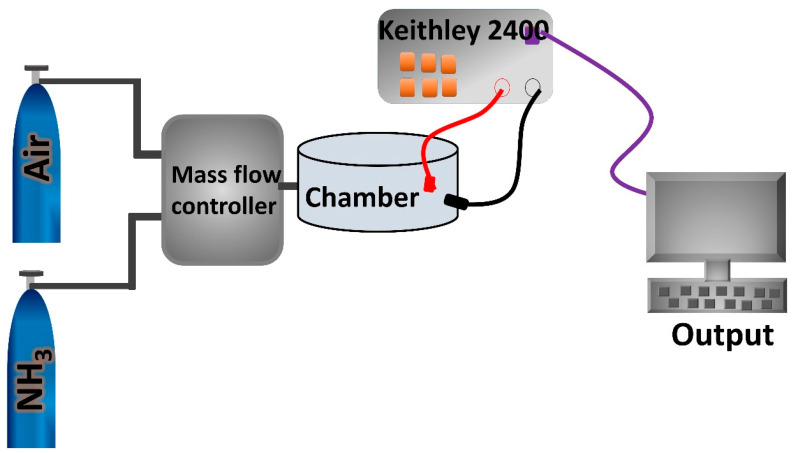
Schematic representation of NH_3_ gas sensor measurement.

**Figure 3 nanomaterials-12-04423-f003:**
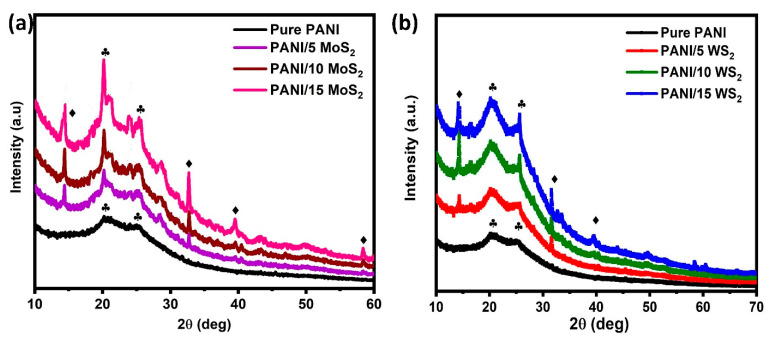
XRD pattern of (**a**) PANI/MoS_2_ nanocomposites and (**b**) PANI/WS_2_ nanocomposites.

**Figure 4 nanomaterials-12-04423-f004:**
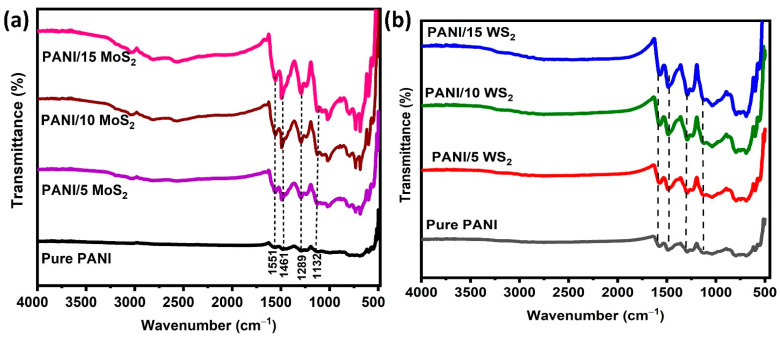
FTIR spectra of (**a**) PANI/MoS_2_ nanocomposites and (**b**) PANI/WS_2_ nanocomposites.

**Figure 5 nanomaterials-12-04423-f005:**
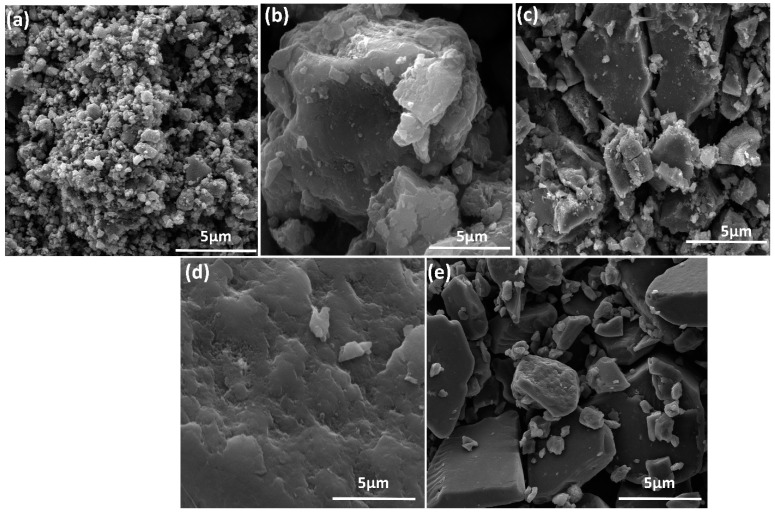
Scanning electron microscope images of (**a**) PANI, (**b**) MoS_2_, (**c**) PANI/MoS_2_ nanocomposites, (**d**) pure WS_2_ and (**e**) PANI/WS_2_ nanocomposites.

**Figure 6 nanomaterials-12-04423-f006:**
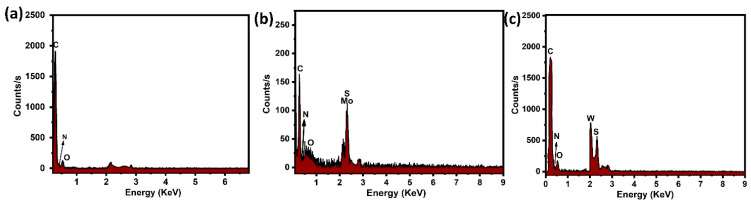
EDX analysis of (**a**) pure PANI, (**b**) PANI/MoS_2_ nanocomposites and (**c**) PANI/WS_2_ nanocomposites.

**Figure 7 nanomaterials-12-04423-f007:**
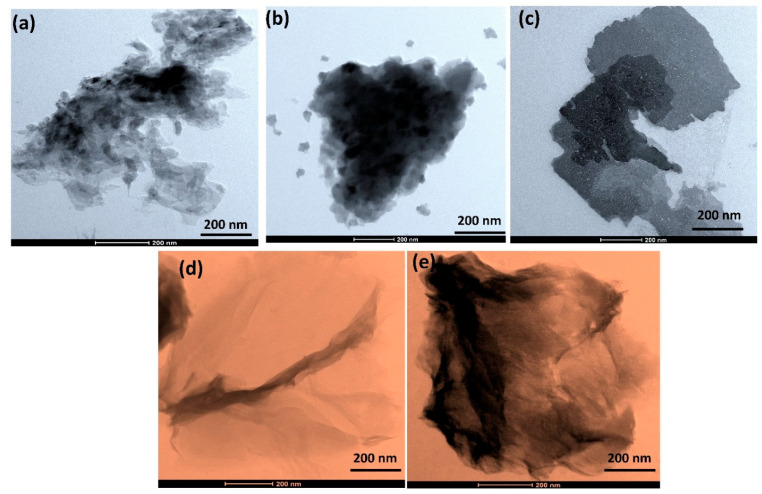
Transmission electron microscope images of (**a**) PANI, (**b**) MoS_2_, (**c**) PANI/MoS_2_ nanocomposites, (**d**) pure WS_2_ and (**e**) PANI/WS_2_ nanocomposites.

**Figure 8 nanomaterials-12-04423-f008:**
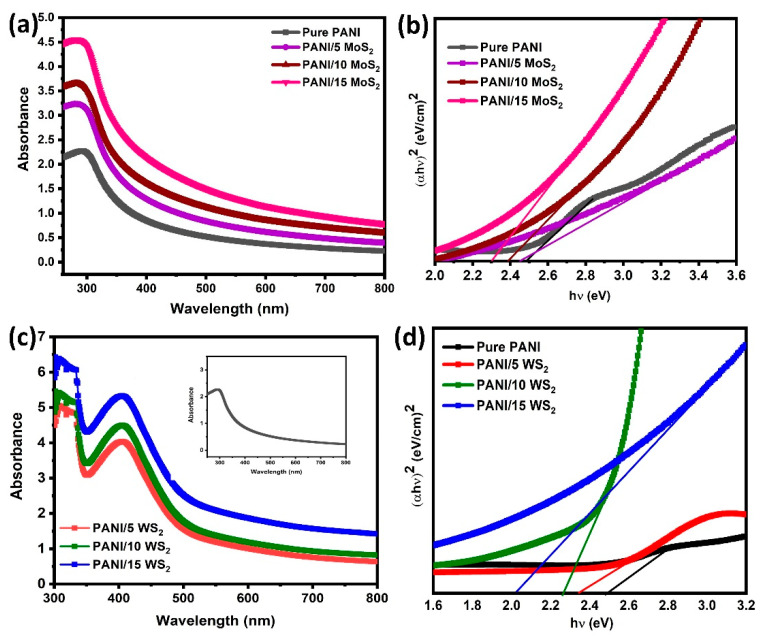
UV-vis absorption spectra and Tauc’s plot of (**a**,**b**) PANI/MoS_2_ composites and (**c**,**d**) PANI/WS_2_ composites.

**Figure 9 nanomaterials-12-04423-f009:**
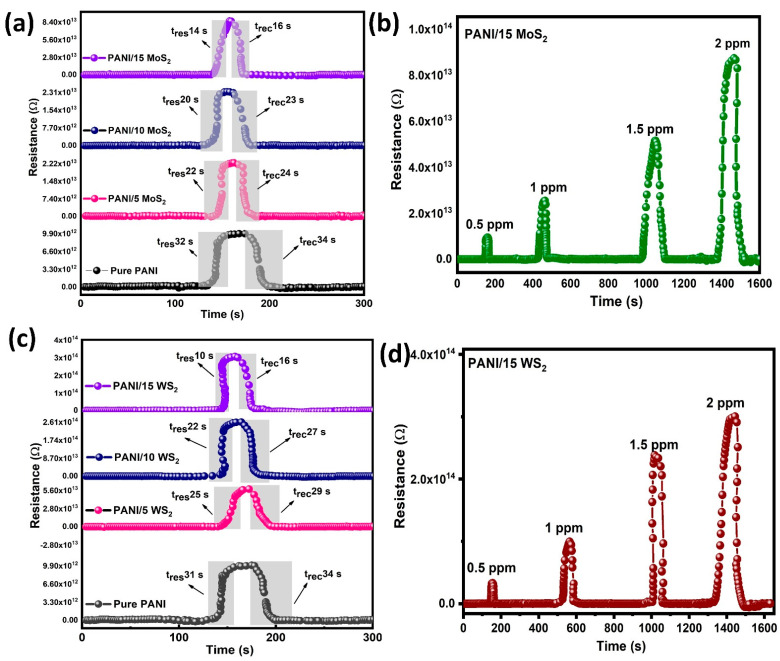
(**a**,**c**) Response and recovery graphs of pure PANI and its composites. (**b**,**d**) Response of PANI and its composites sensor towards NH_3_ gas concentrations (0.5, 1, 1.5 and 2 ppm) operating at room temperature.

**Figure 10 nanomaterials-12-04423-f010:**
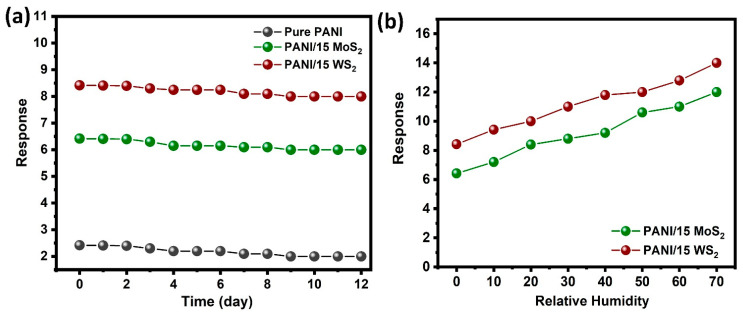
(**a**) Long term stability of the sensors towards 2 ppm NH_3_ at room temperature, (**b**) The influence of humidity on the NH_3_ sensing properties of the sensitive films.

**Figure 11 nanomaterials-12-04423-f011:**
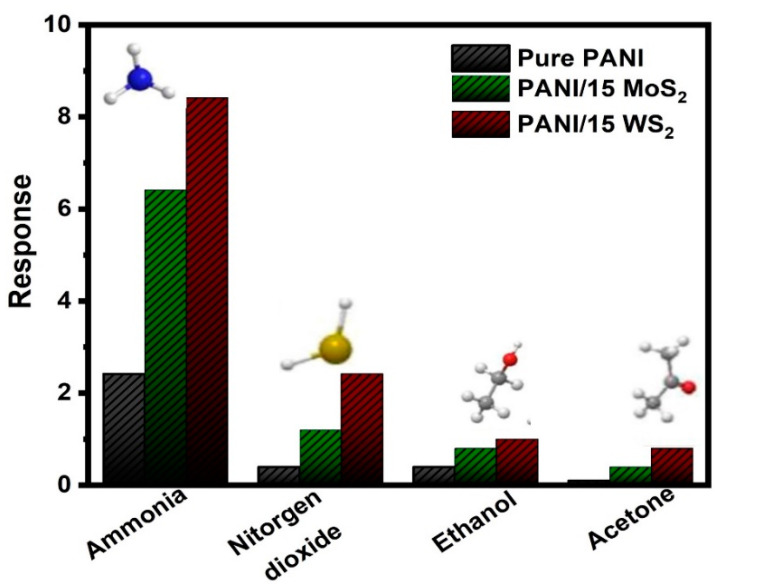
Selectivity test of the gas sensor toward ammonia gas with various gases (Nitrogen dioxide, Ethanol and Acetone).

**Table 1 nanomaterials-12-04423-t001:** The structural and optical parameters for the prepared pure PANI and its nanocomposites.

Samples	Microstrain × 10^−3^	Dislocation Density × 10^14^ Lines/m^2^	Crystallite Size (nm)	Band Gap E_g_ (eV)
Pure PANI	0.8286	5.7145	41.83	2.50
PANI/5 MoS_2_	1.0777	10.6666	22.86	2.45
PANI/10 MoS_2_	1.6613	22.9719	20.16	2.38
PANI/15MoS_2_	2.6730	59.4644	18.96	2.29
PANI/5WS_2_	1.6098	21.1864	21.18	2.35
PANI/10WS_2_	2.0169	33.5701	17.53	2.26
PANI/15WS_2_	2.1420	38.8556	16.18	2.02

**Table 2 nanomaterials-12-04423-t002:** NH_3_ gas-sensing performance of TMD-based systems.

SI. No	Material	Gas	Response Time	Recovery Time	Ref
1	rGO-WS_2_	NH_3_	60 s	300 min	[37]
2	MoS_2_ thin film	NH_3_	22 s	30 s	[38]
3	MoS_2_, WS_2_	NH_3_	5 min	30 min	[39]
4	WS_2_ nanoflakes	NH_3_	120 s	150 s	[40]
5	PANI/MoS_2_/SnO_2_	NH_3_	21 s	130 s	[41]
6	PANI/WS_2_	NH_3_	260 s	790 s	[42]
7	PANI/TMDs	NH_3_	10 s	16 s	This work

## Data Availability

Not applicable.

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
