# Peer review of "Comparative Study on Gas-Sensing Properties of 2D (MoS2, WS2)/PANI Nanocomposites-Based Sensor"

_nanomaterials, 2022, doi:10.3390/nano12244423_

Round 1
Reviewer 1 Report (Previous Reviewer 3)
I accept corrections in the publication and authors' responses to my comments.
Reviewer 2 Report (Previous Reviewer 2)
Accept
Reviewer 3 Report (Previous Reviewer 1)
The revisions make the manuscript better. It looks OK to me.
This manuscript is a resubmission of an earlier submission. The following is a list of the peer review reports and author responses from that submission.
Round 1
Reviewer 1 Report
Major revisions:
The paper presents a PANI/(MoS2, WS2) based RT NH3 sensor. The work looks new and the major results are fine.
1. More introduction to both WS2 and MoS2 based chemiresistive-type NH3 sensors should be done to discuss the advantages and disadvantages of both sensors.
2. The testing procedures of the sensors should be briefly described in the experimental section.
3. Discussions on the sensing mechanisms for the enhanced sensitivity and response/recovery speeds must be discussed in details.
4. Selectivity of the sensors to possible interferents such as ethanol, acetone and NO2 should be examined. Also, the humidity effect should be studied.
5. A long term stability of the sensor should be studied for at least couples of weeks.
6. A comparison table is suggested to compare the sensing properties of the developed PANI/TMDs based NH3 sensors to those reported for TMDs(MoS2, WS2) based ones reported in literature.
Reviewer 2 Report
This paper prepared PANI and PANI/2D-TMDCs by in-situ polymerization method and compared their gas-sensing performance. But the study on gas-sensing was insufficient. More experiment data were needed to support the conclusions. In addition, the Text and Images should be well organized to present the results clearly and highlight the innovation of the research work.
Reviewer 3 Report
I cannot recommend the manuscript entitled " Comparative study on gas sensing properties of 2D (MoS2, WS2)/ PANI nanocomposites based sensor" for publication in the Nanomaterials journal, as in my opinion it contains no new or novel contribution to the state of the art, both regarding nanomaterials and sensors / sensing materials. My main concerns are listed below.
The processes for depositing or decorating PANI via oxidative polymerisation of aniline has been known for three decades and has also been employed in the Authors' earlier works.
The detection of ammonia on sensors similar to the described ones, also with polyaniline, have been reported:
The gas sensor utilizing polyaniline/ MoS2 nanosheets/ SnO2 nanotubes for the room temperature detection of ammonia 10.1016/j.snb.2021.129444
· Jha, Ravindra Kumar, et al. "Ammonia vapour sensing properties of in situ polymerized conducting PANI-nanofiber/WS 2 nanosheet composites." New Journal of Chemistry 42.1 (2018): 735-745.
· Meng, Fanli, et al. "MoS 2-templated porous hollow MoO 3 microspheres for highly selective ammonia sensing via a Lewis acid-base interaction." IEEE Transactions on Industrial Electronics 69.1 (2021): 960-970.
· Ghaleghafi, Elahe, and Mohammad Bagher Rahmani. "Exploring different routes for the synthesis of 2D MoS2/1D PANI nanocomposites and investigating their electrical properties." Physica E: Low-dimensional Systems and Nanostructures 138 (2022): 115128.
· Li, Peidong, et al. "Ultra-Sensitive Nh3 Sensor Based on Pani-Assembled G-C3n4 Nanosheets at Room Temperature." Haoyuan and Liu, Fei and Shi, Junjie and Gao, Xuan-Wen, Ultra-Sensitive Nh3 Sensor Based on Pani-Assembled G-C3n4 Nanosheets at Room Temperature.
· Fatima, Tarab, Samina Husain, and Manika Khanuja. "Superior photocatalytic and electrochemical activity of novel WS2/PANI nanocomposite for the degradation and detection of pollutants: Antibiotic, heavy metal ions, and dyes." Chemical Engineering Journal Advances 12 (2022): 100373.
The sensor proposed by the Authors does not exhibit significantly improved performance compared with the state of the art.
Ad experimental
Please describe how the sensors were fabricated, as the previous works of the Authors are not available in Open Access. Additionally In the cited reference item no. 17, the Authors do not describe their measurement set-up, but refer to yet another of their works. What did "some modifications" of the measurement set-up entail?
Ad XRSD analysys
„Diffraction peaks at 20.5áµ’, and 25.3áµ’ corresponding to (020) and (200) pure PANI 96 crystalline planes, respectively.”
I cannot recommend the manuscript entitled " Comparative study on gas sensing properties of 2D (MoS2, WS2)/ PANI nanocomposites based sensor" for publication in the Nanomaterials journal, as in my opinion it contains no new or novel contribution to the state of the art, both regarding nanomaterials and sensors / sensing materials. My main concerns are listed below.
The processes for depositing or decorating PANI via oxidative polymerisation of aniline has been known for three decades and has also been employed in the Authors' earlier works.
The detection of ammonia on sensors similar to the described ones, also with polyaniline, have been reported:
The gas sensor utilizing polyaniline/ MoS2 nanosheets/ SnO2 nanotubes for the room temperature detection of ammonia 10.1016/j.snb.2021.129444
· Jha, Ravindra Kumar, et al. "Ammonia vapour sensing properties of in situ polymerized conducting PANI-nanofiber/WS 2 nanosheet composites." New Journal of Chemistry 42.1 (2018): 735-745.
· Meng, Fanli, et al. "MoS 2-templated porous hollow MoO 3 microspheres for highly selective ammonia sensing via a Lewis acid-base interaction." IEEE Transactions on Industrial Electronics 69.1 (2021): 960-970.
· Ghaleghafi, Elahe, and Mohammad Bagher Rahmani. "Exploring different routes for the synthesis of 2D MoS2/1D PANI nanocomposites and investigating their electrical properties." Physica E: Low-dimensional Systems and Nanostructures 138 (2022): 115128.
· Li, Peidong, et al. "Ultra-Sensitive Nh3 Sensor Based on Pani-Assembled G-C3n4 Nanosheets at Room Temperature." Haoyuan and Liu, Fei and Shi, Junjie and Gao, Xuan-Wen, Ultra-Sensitive Nh3 Sensor Based on Pani-Assembled G-C3n4 Nanosheets at Room Temperature.
· Fatima, Tarab, Samina Husain, and Manika Khanuja. "Superior photocatalytic and electrochemical activity of novel WS2/PANI nanocomposite for the degradation and detection of pollutants: Antibiotic, heavy metal ions, and dyes." Chemical Engineering Journal Advances 12 (2022): 100373.
The sensor proposed by the Authors does not exhibit significantly improved performance compared with the state of the art.
Ad experimental
Please describe how the sensors were fabricated, as the previous works of the Authors are not available in Open Access. Additionally In the cited reference item no. 17, the Authors do not describe their measurement set-up, but refer to yet another of their works. What did "some modifications" of the measurement set-up entail?
Ad XRSD analysys
„Diffraction peaks at 20.5áµ’, and 25.3áµ’ corresponding to (020) and (200) pure PANI 96 crystalline planes, respectively.”
Poly aniline is an amorphous material, as seen by the XRD diffractograms that contain only two very wide amorphous phase signals, 2Θ = 15÷25o. This reflex corresponds to inter-chain or inter-plane distance.
The crystallite size, calculated according to Scherrer's formula to within this particle size , does not make sense. In addition, the question arises why there is such a significant difference in the values of L, calculated for different diffraction maxima. It is necessary to present diffractograms of the samples. It makes no sense to estimate the value of crystallite size for maxima at large angles, since these reflections have a very weak intensity and the values of the FWHM are determined with a large error. It is also necessary to indicate which software was used to determine the FWHM.
There is no scale in the UV spectra. Absorption units are not given (the adsorbance is not dimensionless). On the c panel - why does a pure PANI have no UV / Vis absorption? Signal at 400 nm shows that we are dealing with PANI oligomers and not PANI polymer. The formation of oligomers is the result of carrying out the synthesis without an excess of oxidant. (A. Proń, F. Genoud, C. Menardo, M. Nechtstchein, Synth. Met., 1988, 24, 193.).
“Due to the fact that PANI film shows a p-type semiconductor characteristic when exposed to gas, the adsorbed NH3 gas molecules donate electrons to the PANI chains after adsorption, which results in a decrease in the carrier concentration of PANI and an increase in resistance”
PANI is an atypical conducting polymer that undergoes doping not only via redox reactions, but most relevantly, via acid-base reactions. In such an acid-base reaction, ammonia will cause dedoping of the polymer. Due to this, the poisoning of PANI-based sensors after exposure to ammonia is a significant problem, particularly in the presence of humidity, particularly so for sensors working at room temperature.
The high hygroscopicity of PANI is also a well-known issue that strongly interferes with the operation of the sensors. The Authors have not presented sensor response results to humidity in the manuscript.